# Measurement instruments for parental stress in the postpartum period: A scoping review

**Anne-Martha Utne Øygarden** [1,2]*, **Rigmor C. Berg**[3,4], **Abdallah Abudayya**[2],
**Kari Glavin**[2], **Benedicte Sørensen Strøm**[2]

**1** Centre of Diakonia and Professional Practice, VID Specialized University, Oslo, Norway, **2** Faculty of Health, VID Specialized University, Oslo, Norway, **3** Reviews and Health Technology Assessments, Norwegian Institute of Public Health, Oslo, Norway, **4** University of Tromsø The Arctic University of Norway, Tromsø og Finnmark, Norway

* anne-martha.oygarden@vid.no

**Citation:** Øygarden A-MU, Berg RC, Abudayya A, Glavin K, Strøm BS (2022) Measurement instruments for parental stress in the postpartum period: A scoping review. PLoS ONE 17(3): e0265616. https://doi.org/10.1371/journal.pone.0265616

**Data Availability Statement:** All relevant data are within the paper.

**Funding:** The authors received no specific funding for this work.

## Abstract

### Background

Parenting stress is a particular type of stress that is conceptualized as a negative psychological response to the numerous obligations associated with raising children. Despite a considerable increase in research on parenting stress, little attention has been given to the ways parenting stress are measured.

### Objectives

This scoping review aimed to provide an overview of available instruments measuring parental stress and to describe their psychometric properties.

### Methods

We conducted a scoping review in accordance with international guidelines for scoping reviews. The main search strategy was searches in seven electronic databases. Pairs of reviewers selected relevant studies based on predetermined inclusion and exclusion criteria. Studies had to report one or more psychometric properties of an instrument measuring stress in parents with children 0–12 months. For each included study, we collected information relevant to the review question, guided by the COnsensus based Standards for the selection of health status Measurement INstruments (COSMIN). Finally, we collated, summarized, and reported the findings descriptively.

### Results

From 2164 unique record, 64 studies from 24 countries were included. They described 15 instruments, of which four were generic and eleven parental-specific self-administered instruments. Only two studies examined parental stress among fathers. Eleven of the studies were validation studies, but they only described seven of the 15 instruments. Internal consistency was the only information provided by 73.4% of the included studies. None of the instruments had information on all measurement properties as per the COSMIN criteria, and

**Competing interests:** The authors have declared that no competing interests exist.

there was no information about measurement error, responsiveness, or interpretability for any of the 15 instruments.

## Discussion

There are presently 15 instruments with some associated psychometric information being used to measure parental stress among parents with young children, but the amount of information on the instruments' psychometric properties is slight. There is a need for further research.

## Introduction

The birth of a child is a joyous event for most parents. However, the postpartum period, which includes the first-year post birth, is also recognized as a period of major transition that can be deeply emotional and associated with considerable distress. Infant care demands and changing social role expectations are factors that are known to increase parents' level of stress [1]. Parenting stress is defined as a particular type of stress and is conceptualized as a negative psychological response to the numerous obligations associated with raising children, and its presence is the rule rather than the exception [2–4].

In recent decades, parenting stress has gained increased importance in clinical practice and research [5]. When conducting searches for a systematic review and using the search terms 'stress' and 'parent', Louie and colleagues [5] found 301 publications in the 1970s, with the number increasing to 4436 publications in the first decade of the 2000s [5]. This reflects a dramatic increase over the last 40 years in research efforts to understand parenting stress. Today, stress is established as an important factor with regard to the well-being of parents, children, and families. However, despite a considerable increase in the number of publications, it seems that little attention has been given to the ways parenting stress are measured, especially in relation to parents with young children [6,7]. The postpartum period is a crucial period in parents' lives [8,9]. It represents a major life transition for most parents [9], while also providing a unique opportunity to screen for stress given parents' regular contact with their public health nurse during the first year.

Previous research has emphasized the need to develop more complex and comprehensive models to examine the effects of different types of stressors in parents [10,11]. Consequently, many parenting interventions measure parental stress levels and a variety of instruments intended to map stress in connection with the parental role, are used [12–17]. A recent review of parenting stress in families where the child was 2–17 years and had clinical issues found that the psychometrics varied across instruments. The combined findings supported the existence of a parenting stress construct and further confirmed the relevance of parenting stress to family functioning, youth psychopathology, and mental health interventions [7]. Another scoping review identified and described interventions for reducing caregiver stress in families where the child suffers from serious illness [6]. The researchers found 49 studies representing six domains of interventions, and a wide variety of measures and standardized questionnaires being used for caregiver stress [6]. To our knowledge, however, there are no reviews providing an overview of available stress measurement instruments, and their psychometric characteristic, used for parents in the postpartum period who have received healthy children. Hence, a scoping review is needed to, one, give a valid overview of existing stress measurements and their psychometric properties, used for parents with children 0 to 12 months; two, facilitate the

choice of an appropriate stress measure fit for purpose; and three, illustrate the gaps and needs in research.

## Objectives

The overall aim of this scoping review is to provide an overview of available instruments measuring parental stress throughout the postpartum period, and describe their psychometric properties related to the relevant population. Our two research questions were: What instruments are available to measure parental stress during the postpartum period? What are the psychometric properties of these instruments?

## Methods

Scoping reviews are used to present a broad overview of the evidence pertaining to a topic, generally with the aim of determining what range of evidence is available and addressing a broader research question [18,19]. We conducted the review in accordance with the five-stage methodological framework proposed by Arksey and O'Malley [19], and further enhanced by Levac, Colquhoun [18]: Specification of the research question, identification of relevant literature, selection of relevant studies, charting data, and collating, summarizing and reporting of results.

### Specification of the research question

We extensively scoped and read existing literature before determining the research question. The research question and a priori methodology was specified in a protocol, registered in Cristin, published in Research Gate [20] and available by contacting the first author.

### Identification of literature

Our main search strategy was searches in electronic databases. The search strategy was developed by the first author and a search specialist. In March 2020, they conducted a systematic search in seven databases: Medline (Ovid), CINAHL, EMBASE (Ovid), Health and Psychosocial Instruments, PsycINFO (Ovid), SveMed+, and Web of Sciences. The search strategy as first formulated in Medline and adapted to the other six databases was:

> 'postnatal' OR 'postpartum care' OR 'perinatal' OR 'postnatal care' OR 'perinatal care' OR 'postpartum' OR 'postpartum period' OR 'postnatal period' AND 'parental stress' OR 'parental distress' OR 'maternal stress' OR 'maternal distress' OR 'Stress+' OR 'Stress, Psychological+' AND 'Psychometrics' OR 'measurement' OR 'Weights and Measures+' OR 'Scales'

> In addition, we conducted a hand search in the reference lists of included publications.

### Selection of relevant studies

We selected relevant studies based on predetermined inclusion and exclusion criteria. We included any study design provided the study measured and reported on stress among parents (mothers and fathers) of children 12 months or younger. We chose the whole first year postpartum, because the first year after birth is a crucial period in parents' lives [8,9], that represents a major life transition for most parents [9]. The study had to report one or more psychometric properties of an instrument to measure stress, understood as described in the introduction. While we only included studies about instruments that measured stress, because

also the word 'distress' is used by some researchers in the field, we included this alternative terminology in the search to ensure that we did not miss relevant studies. In this study, psychometric property was understood as described by the COnsensus based Standards for the selection of health status Measurement INstruments (COSMIN) definition of domains [21], reliability, validity, and responsiveness. The measure of stress had to be undertaken during the postpartum period, which we defined as up to 12 months after birth. Studies had to be published between the years 1995–2020 and written in English or a Scandinavian language. These are the languages mastered by the author team and there were no funds available for study translations.

We excluded studies on parents with seriously ill children (e.g.: cancer, diabetes, preterm), parents who were seriously ill (e.g.: cancer, HIV), parents younger than 18 years, parents with children older than 12 months. Serious illness was defined as *"a health condition that carries a high risk of mortality and either negatively impacts a person's daily function or quality of life or excessively strains the caregiver"* [22].

All records identified in our search were imported into EndNote and duplicates were deleted. We imported all references into Rayyan systematic review software, which is a web-tool designed to help researchers working on scoping reviews and other knowledge syntheses [23]. Using Rayyan, two authors independently screened all titles and abstracts for relevance against the inclusion and exclusion criteria. They promoted all abstracts they considered relevant to full text screening. Having obtained the publications in full text, two independent reviewers assessed their relevance against the inclusion criteria. Studies that met all eligibility criteria were included. At both screening levels, discrepancies or difficulties were deliberated and consensus reached by discussion.

## Charting data

The process of charting data involves applying a common analytical framework to all the included research reports, and collecting on each study standard information relevant to the review question, which is entered onto a data charting form [19]. Using a data extraction sheet (charting table), the first reviewer extracted information, which was checked for accuracy and completeness by another reviewer. The final data extraction sheet was developed after pilot testing it on 11 publications and modifications agreed by two reviewers. We extracted the following data from all publications: year of publication, study setting/country, number of participants, study population characteristics, timepoint of measurement, and study design. Extracted characteristics about the instruments were: instrument name, author, construct(s), target population, method of administration, recall period, (sub)scales, number of items, response options, range of score, and psychometric information. We also extracted data regarding the following measurement properties, based on the COSMIN guidance: internal consistency, reliability, measurement error, content validity, structural validity, hypotheses testing, cross-cultural validity, criterion validity, responsiveness, and interpretability [21]. In accordance with scoping review methodology, we did not perform methodological quality assessments of the included studies [18,19].

## Collating and summarizing results

Finally, in the last step, we collated, summarized, and reported the findings descriptively. We grouped the data into clusters according to instruments and measurement properties, following a data driven approach [19,24]. We described and categorized the psychometric results reported in the studies in accordance with the COSMIN definitions of measurement properties [21]. After conducting descriptive analyses by using frequencies and cross-tabulations, we

recorded findings and discussed implications of the findings. We have reported in accordance with PRISMA Extension for Scoping Reviews [25].

## Results

### Search results

The electronic database searches and the hand searches yielded a total of 9026 records. After deleting duplicates, we screened titles and abstracts. Three publications were not found in full text, and we excluded 198 publications after full text screening. All publications read in full text were in English. A complete list of excluded publications read in full text is available upon request from the corresponding author. We included 64 studies. The study selection procedure is shown in PRISMA Flow Diagram [26], Fig 1.

### Description of included studies

The 64 included studies were all written in English and published between 1999 and 2020, with half being published since 2015 (Table 1). The studies were conducted in twenty-four different countries (Australia, Belgium, Canada, Chile, China, Denmark, Finland, Germany, Ghana/Côte d'Ivoire, Hong Kong, Indonesia, Iran, Israel, Italy, Japan, Lebanon, Norway, Portugal, Spain, Switzerland, Taiwan, Turkey, UK and US). A third of the included studies were conducted in Europe. The sample sizes ranged from 40 to 3005 participants, with a total of 26,783 participants. A third had a sample size greater than 500 participants. Only two studies focused solely on parental stress among fathers, 17 studies looked at both mothers and fathers, while the remaining 45 studies examined stress among mothers. Across the studies, 81 measurements were conducted in the first year postpartum. Most (55 measurements, 68%) were

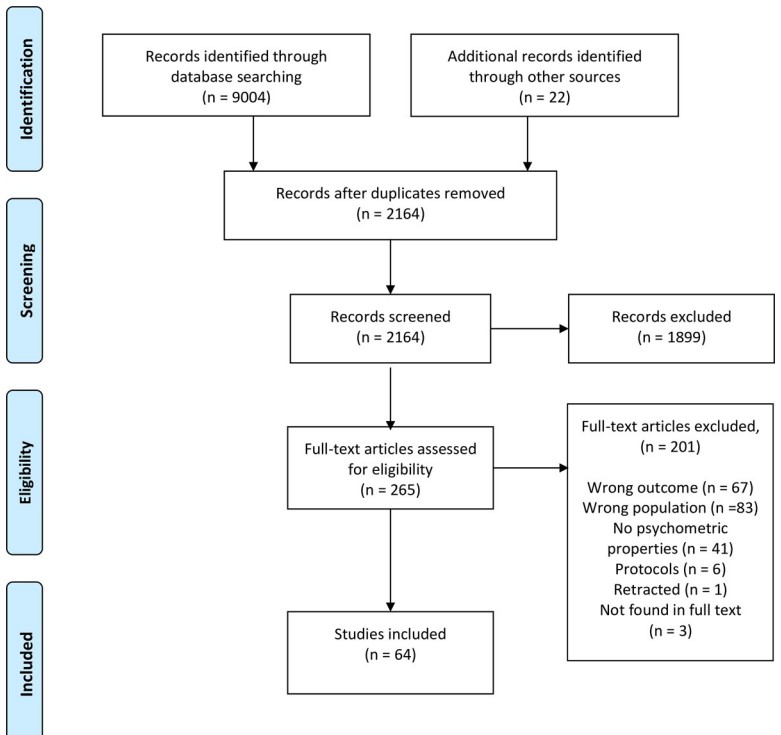

**Fig 1. PRISMA flow diagram.**

**Table 1. Summary characteristics of the included studies (N = 64).**

| Characteristics | Studies N (%) |
| --- | --- |
| **Year of publication:** | |
| 1999–2004 | 6 (9) |
| 2005–2009 | 7 (11) |
| 2010–2014 | 19 (30) |
| 2015–2020 | 32 (50) |
| **Country** | |
| Australia | 3 (4.7) |
| Belgium | 2 (3.1) |
| Canada | 3 (4.7) |
| China | 3 (4.7) |
| Iran | 2 (3.1) |
| Italy | 5 (7.8) |
| Norway | 3 (4.7) |
| Portugal | 3 (4.7) |
| Spain | 3 (4.7) |
| Switzerland | 2 (3.1) |
| Taiwan | 9 (14.1) |
| USA | 14 (21.9) |
| Other (one study from each country) | 12 (18.8) |
| **Number of participants** | |
| <50 | 2 (3) |
| 50–99 | 7 (11) |
| 100–499 | 35 (55) |
| 500–999 | 15 (23) |
| >1000 | 5 (8) |
| **Population** | |
| Fathers | 2 (3) |
| Mothers | 45 (70) |
| Fathers and mothers | 17 (27) |
| **Study design** | |
| Cross-sectional | 24 (37.5) |
| Longitudinal | 21 (33) |
| Validation | 11 (17) |
| Other | 8 (12.5) |
| **Time of measurement of stress [1]** | **N = 81 (%)** |
| Immediately after birth | 8 (9.8) |
| 0–6 months postpartum | 55 (67.9) |
| 7–12 months postpartum | 12 (14.8) |
| Within the first 12 months postpartum | 6 (7.4) |

[1] = more than one answer possible.

conducted 0–6 months postpartum, while eight were conducted immediately after birth and 12 were conducted 7–12 months postpartum. There were different study designs, and 11 of the 64 studies were validation studies, meaning a study that examines the extent to which an instrument measures what it is supposed to measure [27]. These studies validated seven of the 15 identified instruments. Parental stress was assessed both to reveal parents' perspectives and

help to monitor intervention responses. Instruments for parental stress were commonly used in cross-sectional and longitudinal studies as either primary or secondary outcome measures. In cross-sectional studies, parental stress scales were used to evaluate parental stress and determine its relationship with relevant sociodemographic and health-related variables. In longitudinal design, parental stress scales were used to measure changes in parental stress over time as a result of exposure to certain conditions.

## Descriptive characteristics of included instruments

Tables 2 and 3 summarize the characteristics of the included instruments. We identified 15 different instruments measuring parental stress among parents within the postpartum period (children 0 to 12 months). The target population for 11 of the instruments were parents or mothers, while the remaining four instruments were generic stress scales (Perceived Stress Scale, Depression Anxiety Stress Scale, Social Readjustments Rating Scale, Stress Appraisal Measure).

All instruments were self-report scales. The number of instrument items ranged from 4 to 123 (M = 32.03, SD = 28.45). The Perceived Stress Scale (Perceived SS) had the version with the least number of items (4 items), whereas one version of the Parenting Stress Index (PSI) [28] had the most (123 items).

Five of the 15 instruments operated with several item versions (Table 2), with varying degrees of explanation related to the different versions: PSI, Perceived SS, Depression Anxiety

**Table 2. Characteristics of the included instruments.**

| Name of instrument, author (year of development) | Used by how many studies (items-version) | Construct and target population | Recall period | (Sub) scale (s)/ domain (number of items) | Response options and score range |
|---|---|---|---|---|---|
| **Parenting Stress Index (PSI),** Abidin (1982) [28], Abidin (1995) [29] | 4 (PSI-101) | Measure the relative stress in the parent-child relationship. Target population: parents of children age 1 month to 12 years, primary pre-schoolers. | ns | 6 (101) | 1–5; 101–505 |
| | 1 (PSI-123) | | ns | ns (123) | 1–5; 123–615 |
| | 1 (PSI-25) | | ns | ns (25) | 1–5; ns |
| PSI Short Form (PSI-SF), Abidin (2011) [30] | 12 (PSI-SF) | | Past week | 3 (36) | 1–5; 36–180 |
| **Perceived Stress Scale,** Cohen (1983) [31], Cohen (1994) [32], Cohen et.al. (2011) [33] | 6 (PSS-14) | Measure degree to which individuals appraise situations in their lives as unpredictable, uncontrollable, and overloading. Target population: general population. | Last month | ns (10) | 0–4; 0–56 |
| | 9 (PSS-10) | | | (14) | 0–4; 0–40 |
| | 1 (PSS-4) | | | (4) | 0–4; 0–16 |
| **Hung Postpartum Stress Scale (HPSS),** Hung (2005) [34], Hung (2001) [35] | 1 (HPSS-42) | Measure stress in postpartum women. Target population: postpartum women. | During the pureperium | 3 (42) | 1–5; 42–210 |
| | 1 (HPSS-59) | | | ns (59) | ns |
| | 1 (HPSS-62) | | | ns (62) | 1–5; 62–310 |
| | 5 (HPSS-61) | | | ns (61) | 1–5; 61–305 |
| **Parenting Stress Scale (PSS),** Berry and Jones (1995) [36] | 4 (PSS-18) | Developed to capture individual levels of stress associated with raising children. Target population: parents. | The parental period | Unidimensional (18) | 1–5; 18–90 |
| | 1 (PSS-17) | | | (17) | |
| | 1 (PSS-12) | | | (12) | |
| | 1 (PSS-10) | | | (10) | |
| **Depression Anxiety Stress Scale (DASS),** Lovibond and Lovibond (1995) [37] | 3 (DASS-21) | Measure and distinguish between depression and anxiety, and stress. Target population: general population. | ns | 3 (21) | 0–3; 0–63 |
| | 1 (DASS-42) | | Past week | 3 (42) | 0–3; 0–126 |
| **Childcare Stress Inventory,** Cutrona 1983 [38] | 1 | Measure stressful postpartum events of parenthood, specifically related to childcare. Target population: not specified. | Postpartum period | ns (20) | 0–100; 0–2000 |

(*Continued*)

**Table 2.** (Continued)

| Name of instrument, author (year of development) | Used by how many studies (items-version) | Construct and target population | Recall period | (Sub) scale (s)/ domain (number of items) | Response options and score range |
|---|---|---|---|---|---|
| **Postpartum Childcare Stress Checklist**, Dennis et.al. (2018) [39] | 1 | Measure parental perceptions of postpartum childcare stress. Update of CSI [25]. Target population: mothers in early postpartum period. | Early in the postpartum period | ns (19) | ns; 0–23 |
| **Postnatal Perceived Stress Inventory**, Razurel et.al (2013) [40] | 2 | Evaluate interventions for perceived stress during the post-natal period. Target population: mothers. | Postnatal period | 6 (27) | 1–5; 27–135 |
| **Post-delivery perceived stress inventory**, Razurel et.al. (2014) [41] | 1 | Measure post-delivery perceived stress among primiparous women. Target population: primiparous mothers. | From delivery and forward | 6 (29) | 1–5; 29–145 |
| **Cognitive Appraisal of Parenting as Stressful**, Folkman and Lazarus (1985) [42] | 1 | Measure the degree to which parents experienced parenthood, parenting daily hassles, and parenting-related experiences as stressful. Target population: parents. | ns | ns (11) | 1–6; 11–66 |
| **Postpartum Stressors Scale**, Park et.al. (2015) [43] | 1 | Assessment of the type and magnitude of stressors during postpartum period. Target population: postpartum women | Postpartum period | ns (9) | 1–4; 9–36 |
| **Psychosocial Hassles Scale**, Misra et.al. (2001) [44] | 1 | Measure perceived maternal stress due to common stressors. Target population: mothers | Last month | ns (11) | "no stress" to "severe stress"; ns |
| **Rearing-Related Stress**, Sato et.al. (1994) [45] | 1 | Measure mothers' difficulties concerning stress related to child-rearing, conceptualized as a rearing-related stress. Target population: mothers. | ns | 2 (10) | 1–4; 10–40 |
| **Social Readjustments Rating Scale**, Holmes and Rahe (1967) [46]. Also known as **the Holmes and Rahe Stress Scale** | 1 | Measure external stressors. Target population: general population. | Last 12 months | ns (39) (originally 43) | Dichotomous; ns |
| **Stress Appraisal Measure**, Peacock (1990) [47] | 1 | Measure an individual's appraisal of a specific stress situation identified by the examiner. Target population: general population. | ns | 7 (28) | 1–5; 28–140 |

Note: Table based on COSMIN standard [21]. ns = not stated.

Stress Scale (DASS), Hung Postpartum Stress Scale (HPSS), and Parenting Stress Scale (PSS). First, the PSI [28,29] is originally a 101-items instrument, with an optional 19-item Life Stress scale. We included four studies, but no validation studies, of the original PSI 101-items version conducted on parents with healthy children between 0–12 months [38,48–50]. They found Cronbach's alpha ranging between 0.70–0.94. Two studies used respectively a 25-items version [51] and a 123-items version [17]. The 25-items version is claimed to be based on the original 101-item [51], but it is unclear which items constitute this version. In the 123-items version, the optional Life Stress dimension consists of 22-items [17]. PSI also exists with a widely used 36-item short version [29]. We included one validation study of PSI-Short Form (PSI-SF) [62], and 11 other studies used this version. Cronbach's alpha in these studies ranged between 0.77–0.96. The extensively used Perceived SS [31] is originally a 14-items instrument, with seven positive- and seven negative items. Later, a 10-items version was introduced, and we included one validation study of Perceived SS-10 [71], with a Cronbach's alpha for postpartum women of 0.71. There also exists a short 4-items version that can be made from questions 2, 4, 5 and 10 of the Perceived SS-10 version [101]. One study in this sample used the 4-items version [102]. DASS [37] exists primarily with 42 items. DASS consists of three individual scales (depression,

anxiety, stress) and each of the three DASS scales contains 14 items, which are divided into subscales of 2–5 items with similar content. A short version of DASS with 21 items also exists, with three scales that each consists of 7 items. Further, in this scoping review we found several versions of HPSS, presented by the same author group. Different item-versions are validated and adjusted in accordance with new developments. Versions vary from 42–62 items [34,35,80,82]. PSS [36] was originally developed as an 18-items scale, but three validation studies conducted on our parent group of interest recommend exclusion of one or more items [85,86,88]. Thus, among our included studies we find PSS with 18, 17, 12, and 10 items. Lastly, we mention that regarding the Psychosocial Hassles Scale [44], Kinsey, Baptiste-Roberts [97] write that they modified several items to be more appropriate for the study population and added an item, resulting in 12 items. They provide no further explanations.

Only six studies [57–59,90,93,98], using respectively DASS, PSI-SF, and Rearing-Related Stress, had a cut-off to indicate threshold of high-level stress. No other studies made any statements regarding cut-off, beyond some stating that 'higher score indicates higher perception of parenting stress'.

## Reported psychometric properties of included instruments

An overview of the psychometric properties for the 15 instruments is presented in Table 3 with references to all included studies. Among the 64 studies describing the 15 instruments, there were 11 validation studies (see Table 3). These 11 studies validated only seven of the 15 instruments found in relation to the targeted group for this scoping review, including one generic stress measure (Perceived Stress Scale). Hence, eight instruments were used, despite not being validated for use, on parents within the postpartum period.

None of the eleven validation studies presented on all ten psychometric properties according to COSMIN. We found no studies that assessed measurement error, responsiveness or interpretability (these properties are therefore not shown in Table 3). The instrument with the highest number of reported psychometric properties was PSS [36]. Information about internal consistency was provided by all 64 studies, except four. These four studies provided the reliability coefficient or content validity. Internal consistency was reported as Cronbach's alpha (α), except for one study that reported McDonald's omega (Ω). Twelve studies reported on construct validity. Five studies assessed criterion validity. Four studies contained information on content validity, using either a group of experts or a working group of patients (face validity). Structural validity was assessed mainly by exploratory or confirmatory factor analysis. Rasch analysis was less common and used only once.

## Discussion

This systematic scoping review had two aims: to provide an overview of available instruments on parental stress throughout the postpartum period, and to report psychometric properties measured related to the relevant population. We included and extracted data from 64 studies reporting on 15 instruments used to assess stress among parents with healthy children 0–12 months.

As per our first objective, we identified four generic and eleven parental-specific self-administered instruments used to assess parental stress among parents with children who were 12 months or younger. There is a visible increase in studies measuring parental stress from 2010 and forward, indicating an increased focus on parental stress as an important factor regarding the well-being of families in addition to already established factors like postpartum depression symptoms. This is in line with the increase also found in other studies [5]. Yet, this increase is geographically skewed, with scant research conducted in South America, Africa,

**Table 3. Overview of psychometric properties for instruments presented.**

| Instruments | Authors (Year) | Psychometric properties | | | | | | |
|---|---|---|---|---|---|---|---|---|
| | | Reliability | | Validity | | | | |
| | | | | Content validity | Construct validity | | | Criterion validity |
| | | A | B | D | E | F | G | H |
| **Parenting Stress Index (PSI)** | Glavin, Smith [17], Gameiro [48], Fredriksen [49], Colpin [50], Anhalt [51] | ✓ | | | | | | |
| | Krieg [52] | | ✓ | | | | | |
| **Parenting Stress Index—Short Form (PSI-SF)** | Lutenbacher [53] | | ✓ | | | | | |
| | Alves, Milek [54]; Canzi [55]; Goodman [56]; McCarter [12], Guo [57] Prino [58], Vismara [59], Molgora [60], Rollè [61] | ✓ | | | | | | |
| | *Aracena [62] | ✓ | | | ✓ | ✓ | ✓ | ✓ |
| | Vanska [63] | ✓ | ✓ | | | | | |
| **Perceived Stress Scale (Perceived SS)** | | | | | | | | |
| **Perceived SS-10** | Walker [16], Gao [64], Gill and Loh [65], Ko [66], Koletzko [67], Lee [68], Mao [69] | ✓ | | | | | | |
| | Mann [70] | ✓ | ✓ | | | | | |
| | *Chaaya [71] | ✓ | | | | ✓ | ✓ | ✓ |
| **Perceived SS-14** | Roman [14], Duran [72], Lu [73], Ramirez [74], Tavares [75], Thorp [76] | ✓ | | | | | | |
| **Perceived SS-4** | Rodriguez [77] | ✓ | | | | | | |
| **Hung Postpartum Stress Scale (HPSS)** | Fang and Hung [78] | ✓ | | | | | | |
| | Hsien [79] | ✓ | | | | | | |
| | *Hung [80] | ✓ | | | ✓ | | | |
| | Lee and Hung [81] | ✓ | | | | | | |
| | *Hung [35] | ✓ | | | ✓ | ✓ | | |
| | *Hung [34] | ✓ | | ✓ | ✓ | ✓ | | |
| | Hung and Chung [82] | ✓ | | | | ✓ | | |
| | Navidian [13] | ✓ | | | | | | |
| **Parenting Stress Scale (PSS)** | | | | | | | | |
| **PSS-18** | Da Costa [83], Unternaehrer [84] | ✓ | | | | | | |
| | *Nærde and Sommer Hukkelberg [85] | ✓ | ✓ | | ✓ | ✓ | | ✓ |
| | *Pontoppidan [86] | | ✓ | ✓ | ✓ | ✓ | ✓ | |
| **PSS-17** | Oltra-Benavent [87] | ✓ | | | | | | |
| **PSS-12** | *Oronoz [88] | ✓ | | | ✓ | | ✓ | ✓ |
| **PSS-10** | Fernandes [89] | ✓ | | | | | | |
| **Depression Anxiety Stress Scale (DASS)** | | | | | | | | |
| **DASS-21** | Wilson [90], Giallo [91], Schwab-Reese [92] | ✓ | | | | | | |
| **DASS-42** | Gillis [93] | ✓ | | | | | | |
| **Childcare Stress Inventory (CSI)/ Postpartum Childcare Stress Checklist** | Nurbaeti [94] | ✓ | | | | | | |
| | *Dennis [39] | ✓ | | ✓ | ✓ | ✓ | | |
| **Postnatal Perceived Stress Inventory** | * Razurel [40] | ✓ | | | | ✓ | | |
| | Tabrizi and Nournezhad [95] | | | ✓ | | | | |
| **Post-delivery perceived stress inventory** | *Razurel [41] | ✓ | | | | ✓ | | |
| **Cognitive Appraisal of Parenting as Stressful** | Levy-Shiff [96] | ✓ | | | | | | |
| **Postpartum Stressors Scale** | Park [43] | ✓ | | | | | | |
| **Psychosocial Hassles Scale** | Kinsey [97] | ✓ | | | | | | |

(*Continued*)

**Table 3.** (Continued)

| Instruments | Authors (Year) | Psychometric properties | | | | | |
|---|---|---|---|---|---|---|---|
| | | Reliability | Validity | | | | |
| | | | | Content validity | Construct validity | | Criterion validity |
| **Rearing-Related Stress** | Cheng [98] | √ | | | | | |
| **Social Readjustments Rating Scale** | Ngai and Chan [99] | √ | | | | | |
| **Stress Appraisal Measure** | Honey [100] | √ | | | | | |

COSMIN psychometric property boxes: A = internal consistency, B = reliability, D = content validity, E = structural validity, F = hypotheses testing, G = cross-cultural validity, H = criterion validity. C = measurement error, I = responsiveness and J = interpretability was removed due to lack of measurement.
*Validation study.

and the Middle East. Only eight percent of the studies were from these regions. This is disconcerting, given the importance of instruments' cross-cultural validity. Related, acknowledging the differences in mothers' and fathers' postpartum stress symptoms, there is a gap in knowledge about measures for fathers' postpartum stress [63,85,86]. The majority of the included studies focused on mothers (70%), while only two (3%) focused solely on fathers [63,83]. Only two of the 11 validation studies included both fathers and mothers [85,88], both validating the Parental Stress Scale [36]. Together, the two validation studies provide psychometric data on internal consistency, reliability, structural validity, hypotheses testing, cross-cultural validity, and criterion validity. While future studies may usefully build upon this work, there remains an uncertainty regarding the instruments' sensitivity to identify fathers' stress level. When selecting the most appropriate instrument for a particular purpose, it is relevant to compare the conceptual and psychometric properties of the pre-existing instruments [103], and take this information into consideration when making the selection [104]. Our scoping review reveals that those interested in assessing parental stress during the postpartum period among parents, and particularly among fathers, outside of North-America, Europe and a few Asian countries have insufficient information to make an evidence-informed decision on which instrument to use.

We found that the majority of the included studies measured parental stress between 0–6 months after birth. Furthermore, seven of the 11 validation studies assessed parental stress as early as the first two months postpartum [34,35,39–41,62,80], while the remaining four studies assessed parental stress when the child was between 6–12 months [71,85,86,88]. Although the earliest postpartum months give a unique opportunity to measure parental stress given parents' frequent contact with their public health nurse, we must be cognizant that the dynamics related to parenting stress change across the first year after birth [9,59,82]. More studies should validate parental stress instruments throughout the first year after birth, as there is limited information on instruments' properties from especially months 3–12, and it would be important to gain more empirical data on the accuracy of instruments throughout the whole first year of transformation to parenthood.

For five instruments, we identified different item versions. In some studies, a rationale for the selected version was missing, as well as which items had been deselected or added. We encourage future researchers to be more transparent in their reporting. We also found a substantial difference in the number of scale items, from 4 items to 123 items. Clearly, the labour intensity for participants will be dramatically different depending on which scale is used, but given the sparse data provided on the instruments' properties the value of this aspect is presently unclear. In addition, only six studies indicated a cut-off score to divide high-level stress

from lower. This is problematic, because proper interpretation of scores is imperative and best facilitated when the instrument developers establish cut-points for classification purposes [105]. Both the difference in number of items and the lack of cut-off score may cause difficulties in selecting instruments, interpreting scores, and comparing results across studies. The most frequently used instrument in this review was a generic instrument, the Perceived Stress Scale, which is said to be the most widely used instrument for measuring the perception of stress [32]. However, we found that only the Arabic version is validated for parents with young children [71]. Although this instrument may measure the burden of stress among parents, it may lack sensitivity in identifying parent-specific problems and can provide misleading results. In addition, using an instrument that is parent-specific may be more effective in identifying parent-related symptoms and problems, and their impacts on the parent-child dyad.

As per our second objective, we documented that for none of the instruments is there information on all their measurement properties as per the COSMIN criteria. Unsurprisingly, the 11 validation studies presented most of the psychometric information about the instruments. Beyond internal consistency, which was the only information provided by 73.4% of the 64 included studies, we only learn of the psychometric properties of seven of the 15 instruments, and there is no information about measurement error, responsiveness, interpretability. Responsiveness is measured to detect changes over time properly. When unmeasured, researchers and healthcare professionals are poorly equipped to use the instrument as an indicator of quality of care in clinical practice and research. Similarly, interpretability is a meaningful requisite for the applicability of instruments in research [106], but also the evidence on interpretability of all 15 instruments is unknown. The instruments with the most comprehensive psychometric assessments are the Parenting Stress Scale, Parenting Stress Index Short Form, and Hung Postpartum Stress Scale, although the latter is assessed only among parents in Asia. For these three, it would be possible to conduct a systematic review including methodological quality assessment of included validation studies, in accordance with the COSMIN guidelines. Researchers interested in measuring parental stress may wish to examine these instruments in particular to select an appropriate stress measure fit for their purpose. Our results mirror those of Holly and colleagues [7], who found a variety of psychometrics across instruments for parenting stress. They concluded that one must consider both the purpose for which the instrument will be used, and the evidence base for the measure when selecting an instrument, as the importance of psychometric categories may vary depending on the purpose of the parenting stress assessment. It is important to stress that we assessed the extent to which the measurement properties of instruments for parental stress have been evaluated, finding that few to none have been thoroughly evaluated. Thus, we find that there is still insufficient evidence to endorse one specific instrument for parental stress measurement.

## Strengths and limitations

The strengths of our scoping review include the systematic searches, selection, and data extraction by two reviewers, and quality assured collation of data. However, in line with scoping review methodology, we conducted no methodological quality assessment of included studies. We also limited the number of languages and had no extensive searches in grey literature sources. Lastly, psychometric properties are reported, not evaluated, which would be important in future research.

## Conclusion

There are presently 15 instruments with some associated psychometric information being used to measure parental stress among parents with young children, but the amount of

information on the instruments' psychometric properties is slight. While internal consistency is known for all 15 scales, their validity, responsiveness, and interpretability are mostly unknown. We find that there is still insufficient data to recommend one parental stress instrument over another, and further research is warranted. The lack of evidence of the accuracy of parenting stress measures makes it challenging to understand and mitigate information bias related to parenting stress, and there is a need for further research on the instruments' measurement properties, in different cultural and language contexts, particularly among fathers.

## Supporting information

**S1 Table. PRISMA-ScR.**
(DOCX)

## Acknowledgments

We are grateful to librarian Irjall for her excellent support and recommendations during the search process.

## Author Contributions

**Conceptualization:** Anne-Martha Utne Øygarden, Rigmor C. Berg.

**Data curation:** Anne-Martha Utne Øygarden.

**Formal analysis:** Anne-Martha Utne Øygarden, Rigmor C. Berg, Abdallah Abudayya, Kari Glavin, Benedicte Sørensen Strøm.

**Investigation:** Anne-Martha Utne Øygarden, Rigmor C. Berg.

**Methodology:** Anne-Martha Utne Øygarden.

**Project administration:** Anne-Martha Utne Øygarden.

**Supervision:** Rigmor C. Berg, Benedicte Sørensen Strøm.

**Visualization:** Anne-Martha Utne Øygarden, Rigmor C. Berg, Benedicte Sørensen Strøm.

**Writing – original draft:** Anne-Martha Utne Øygarden.

**Writing – review & editing:** Anne-Martha Utne Øygarden, Rigmor C. Berg, Abdallah Abudayya, Kari Glavin, Benedicte Sørensen Strøm.

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
