## [Decision Letter · Decision Letter 0]

4 Dec 2021

PONE-D-21-27014Measurement instruments for parental stress in the postpartum period: A scoping reviewPLOS ONE

Dear Dr. Øygarden,

Thank you for submitting your manuscript to PLOS ONE. After careful consideration, we feel that it has merit but does not fully meet PLOS ONE’s publication criteria as it currently stands. Therefore, we invite you to submit a revised version of the manuscript that addresses the points raised during the review process.

Please address all the comments pointed out by the reviewers. You will find that the comments are useful to improve the manuscript.

We look forward to receiving your revised manuscript.

Kind regards,

Kenta Matsumura

Academic Editor

PLOS ONE

Journal Requirements:

Reviewers' comments:

Reviewer's Responses to Questions

**Comments to the Author**

1. Is the manuscript technically sound, and do the data support the conclusions?

Reviewer #1: Partly

Reviewer #2: Partly

2. Has the statistical analysis been performed appropriately and rigorously? 

Reviewer #1: N/A

Reviewer #2: N/A

3. Have the authors made all data underlying the findings in their manuscript fully available?

Reviewer #1: Yes

Reviewer #2: Yes

4. Is the manuscript presented in an intelligible fashion and written in standard English?

Reviewer #1: Yes

Reviewer #2: No

5. Review Comments to the Author

Reviewer #1: p.6 1)Selection of relevant studies: Authors mentioned that although alternative terminology such as ‘distress’ was accepted if the instrument measured stress. Is it appropriate? Please explain this reasons. In addition, authors selected literature written in English and Scandinavian language. Please specify which literature was in Scandinavian language in the Table.

p.13-14 Table2 The title and content in the right column of Table 2 didn't match well. Title of this column is "Name of instrument, author(year of development), but the author name and development year are not written in the one.

Reviewer #2: Introduction:

-Please provide rationale that the author selected 1 year after birth as a focus of the study. First few months is very different from the latter half of the first year.

-objectives stated in the introduction does not reflect conducted analysis-facilitating the choice of an appropriate stress measure fit for purpose. It was not analyzed or reported in the manuscript.

Methods

-Selection of relevant studies: children with seriously ill children. What is the definition of serious illness?

Results

-search results: Please provide information on how many studies were excluded with what reasons. It reported that 3 were excluded due to lack of full text, however, no reasons of excluding other 204 were not provided. Please revise the flow diagram accordingly.

Discussion

-p19 "stress changes across the first year after birth"

Please provide reference to support this statement.

-"there exists insufficient evidence to endorse one specific instrument for parental stress measurement"

I think this is not the purpose of the author. In the introduction section, it was mentioned that authors wanted to facilitate appropriate choice of instrument to fit the purpose. This also applies to the conclusion section.

Conclusion

-Also, according to the discussion, there is insufficient data for measuring stress during the first year of parenting.

Minor comment

-Identification of literature: I addition>In addition

6. PLOS authors have the option to publish the peer review history of their article (what does this mean?). If published, this will include your full peer review and any attached files.

Reviewer #1: No

Reviewer #2: No

---

## [Author Response · Author response to Decision Letter 0]

26 Jan 2022

Thank you for the many constructive suggestions for improvements. For your information, a tidier response is uploaded in the appendix "Response to Reviewers". Here we respond with green text to separate reviewers' feedback from our responses.

Response to Reviewers.

Templates are followed. Mistakes in file naming are corrected. 

All references used are provided in the manuscript. Data in this manuscript is solely articles found in databases and are available through literature search. We apologize for the misunderstandings. 

Captions for Supporting Information files are included at the end of the manuscript.  

Comments to the Author

1. Is the manuscript technically sound, and do the data support the conclusions?

Reviewer #1: Partly

Reviewer #2: Partly

The article is reworked, and the comments and suggestions from the reviewers and the editor have been followed. 

4. Is the manuscript presented in an intelligible fashion and written in standard English?

Reviewer #1: Yes

Reviewer #2: No

Multiple people have carefully read and revised the manuscript. In addition, it is reviewed and proof read by an English language expert. We believe that our paper is presented in an intelligible fashion and written in standard, correct English. Should the editorial team detect any errors or want any other changes, we are of course happy to oblige. ________________________________________

 

Reviewer #1: 

p.6 1) Selection of relevant studies: Authors mentioned that although alternative terminology such as ‘distress’ was accepted if the instrument measured stress. Is it appropriate? Please explain these reasons. 

Reasons explained on p. 7 in the manuscript: “While we only included studies about instruments that measured stress, because also the word ‘distress’ is used by some researchers in the field, we included this alternative terminology in the search to ensure that we did not miss relevant studies.”

Explanation: Our knowledge of the research field has taught us that researchers sometimes use the word ‘distress’ and sometimes the word ‘stress’. This is why the term ‘distress’ was included in the search. 

In addition, authors selected literature written in English and Scandinavian language. Please specify which literature was in Scandinavian language in the Table.

Specified in the result chapter, p. 9: “Three publications were not found in full text and we excluded 198 publications after full text screening. All publications read in full text were in English.” 

Explanation: While we accepted reports in English and Scandinavian languages, no publications in a Scandinavian language met the criteria for full text screening or for inclusion.

p.13-14 Table2 The title and content in the right column of Table 2 didn't match well. Title of this column is "Name of instrument, author (year of development), but the author name and development year are not written in the one.

Corrected in Table 2. 

Explanation: Author and year was moved to numbers as a result of the Vancouver style. Corrected manually. 

Reviewer #2: 

Introduction:

-Please provide rationale that the author selected 1 year after birth as a focus of the study. First few months is very different from the latter half of the first year.

Rationale provided on p. 7: “We chose the whole first year postpartum, because the first year after birth is a crucial period in parents’ lives [8, 9], that represents a major life transition for most parents [9].”

Explanation: We agree that there are likely differences in parental distress between the first few months after birth and the last half of the first year, and we recognize that there are different definitions for the postpartum period. We have provided references to further substantiate the claim that the first year after the birth of a child is a crucial period and major life transition for parents. However, we wanted to be inclusive of the first year after birth, and to map broadly, and therefore included the entire first year after birth. We provide differentiation of measurement times in Table 1 (and text in the result section), allowing readers to see and focus on when measurements have been used to map stress among parents. Readers who are specifically interested in the first few months after birth can focus on those studies.

-objectives stated in the introduction does not reflect conducted analysis-facilitating the choice of an appropriate stress measure fit for purpose. It was not analyzed or reported in the manuscript.

Changes made in result (p. 10), discussion (p. 19) and conclusion (p. 22) chapter.

Explanation: There are two objectives in this scoping review stated under objectives, p. 5, this is: [to] provide an overview of available instruments measuring parental stress throughout the postpartum period, and describe their psychometric properties related to the relevant population. In the comment above, reviewer 2 seems to refer to the introduction where it is stated that a scoping review is needed to, among other things, facilitate the choice of an appropriate stress measure fit for purpose. This is still needed. However, we re-examined our presentation of result, discussion and conclusion and made some changes that we believe make the presentation clearer. 

Methods

-Selection of relevant studies: children with seriously ill children. What is the definition of serious illness?

Definition provided on p. 7: Serious illness was defined as “a health condition that carries a high risk of mortality and either negatively impacts a person’s daily function or quality of life or excessively strains the caregiver.” (Kelley and Bollens-Lund, 2018). 

Explanation: We excluded a few articles where measurement was done among parents having an infant at the neonatal infant care unit or where one or two parents were living with HIV. These articles are sorted under wrong population in the PRISMA Flow Diagram, S1_fig.tif

Results

-search results: Please provide information on how many studies were excluded with what reasons. It reported that 3 were excluded due to lack of full text, however, no reasons of excluding other 204 were not provided. Please revise the flow diagram accordingly.

The three articles not found in full text was included in the 201 articles excluded. Therefore, three publications were not found in full text and 198 publications were excluded after full text screening. 

Reasons for 198 articles excluded + 3 articles not found are now provided in the PRISMA Flow Diagram, see Fig1.tif. We hope this make the presentation clearer.

Discussion

-p19 "stress changes across the first year after birth"

Please provide reference to support this statement.

Two reference provided on p. 20: [9, 43, 54] 

-"there exists insufficient evidence to endorse one specific instrument for parental stress measurement"

I think this is not the purpose of the author. In the introduction section, it was mentioned that authors wanted to facilitate appropriate choice of instrument to fit the purpose. This also applies to the conclusion section.

Changed on p. 22: Thus, we find that there is still insufficient evidence to endorse one specific instrument for parental stress measurement.

Explanation: It was stated in the introduction that a scoping review is needed to, among other things, facilitate the choice of an appropriate stress measure fit for purpose. This last sentence in the discussion is aiming to point towards what research we need in the future. 

Conclusion

-Also, according to the discussion, there is insufficient data for measuring stress during the first year of parenting.

Changed on p. 23: We find that there is still insufficient data to recommend one parental stress instrument over another, and further research is warranted.

Minor comment

-Identification of literature: I addition>In addition

Corrected on p. 6. 

6. PLOS authors have the option to publish the peer review history of their article. If published, this will include your full peer review and any attached files.

We would not like to publish the peer review history of this article.

---

## [Decision Letter · Decision Letter 1]

7 Mar 2022

Measurement instruments for parental stress in the postpartum period: A scoping review

PONE-D-21-27014R1

Dear Dr. Øygarden,

We’re pleased to inform you that your manuscript has been judged scientifically suitable for publication and will be formally accepted for publication once it meets all outstanding technical requirements.

Kind regards,

Kenta Matsumura

Academic Editor

PLOS ONE

Additional Editor Comments (optional):

Thank you for your excellent work!

Reviewers' comments:

Reviewer's Responses to Questions

**Comments to the Author**

1. If the authors have adequately addressed your comments raised in a previous round of review and you feel that this manuscript is now acceptable for publication, you may indicate that here to bypass the “Comments to the Author” section, enter your conflict of interest statement in the “Confidential to Editor” section, and submit your "Accept" recommendation.

Reviewer #1: (No Response)

Reviewer #2: All comments have been addressed

Reviewer #3: All comments have been addressed

2. Is the manuscript technically sound, and do the data support the conclusions?

Reviewer #1: Yes

Reviewer #2: Yes

Reviewer #3: Yes

3. Has the statistical analysis been performed appropriately and rigorously? 

Reviewer #1: Yes

Reviewer #2: N/A

Reviewer #3: Yes

4. Have the authors made all data underlying the findings in their manuscript fully available?

Reviewer #1: (No Response)

Reviewer #2: Yes

Reviewer #3: Yes

5. Is the manuscript presented in an intelligible fashion and written in standard English?

Reviewer #1: Yes

Reviewer #2: Yes

Reviewer #3: Yes

6. Review Comments to the Author

Reviewer #1: Thank you for your research with valuable information. Parenting stress is a very important issue of Nursing, and the development of research using appropriate scales is important for children and families.

Reviewer #2: (No Response)

Reviewer #3: The authors have sufficiently revised the manuscript, answering all comments that the reviewers provided. The manuscript is worthwhile considering to be accepted.

7. PLOS authors have the option to publish the peer review history of their article (what does this mean?). If published, this will include your full peer review and any attached files.

Reviewer #1: No

Reviewer #2: No

Reviewer #3: No

---

## [Editor Report · Acceptance letter]

11 Mar 2022

PONE-D-21-27014R1 

Measurement instruments for parental stress in the postpartum period: A scoping review 

Dear Dr. Øygarden:

I'm pleased to inform you that your manuscript has been deemed suitable for publication in PLOS ONE. Congratulations! Your manuscript is now with our production department. 

Kind regards, 

on behalf of

Dr. Kenta Matsumura 

Academic Editor

PLOS ONE